# UDIS: A User-query Driven Framework for Image Forgery Localization

## Abstract

The rapid advancement of image editing technologies has amplified the urgency of developing reliable Image Forgery Localization (IFL) methods. Recent approaches based on Multimodal Large Language Models (MLLMs) have shown promise but suffer from **weak visual-text alignment**: they fail to regulate visual attention to the specific regions mentioned in user queries, leading to irrelevant responses. We argue that this limitation originates from a **global outcome driven** paradigm that directs interpretability toward forgery localization results and focuses visual attention on the entire image. To address this issue, we propose a paradigm shift: interpretability in IFL ought to be **regional user-query driven**. Building on this principle and supported by a dataset containing queries related to the authenticity of specific regions, we present the **U**ser-query **D**riven **I**mage **S**hield (UDIS), a novel framework incorporating two key modules. The **Query-Guided Module (QGM)** introduces a `[QUERY]` token and a visual features filtering process based on the queries to strengthen the **input-level** alignment (focusing on connecting query and MLLM's visual attention). The **Evidence-Aware Module (EAM)** introduces an `[EVI]` token and an auxiliary authenticity evidence classification task to enhance alignment at the **output-level** (focusing on associating explanatory text knowledge with forgery localization capability). By learning the two special tokens, MLLM's alignment ability is enhanced, and the modal-consistency knowledge embedded in the tokens further supports the forgery localization process. Extensive experiments demonstrate that the proposed approach provides query-focused authenticity explanations, underscoring its superior practical value, and achieves state-of-the-art IFL performance.

## 1 Introduction

With the rapid advancement of image editing technologies, their potential risks have raised widespread concerns, underscoring the urgent need to develop effective Image Forgery Localization (IFL) methods to safeguard trust and security in public spaces. In recent years, researchers have developed various IFL methods based on Multimodal Large Language Models (MLLMs) (Liu et al., 2024; Xu et al., 2024; Huang et al., 2025; Yang et al., 2025; Kang et al., 2025). As shown in Figure 1, existing approaches typically take a tampered image and a forensics-related question as input, and output a forgery localization result, together with explanatory text describing why the regions are tampered. However, these methods often suffer from **weak visual-text alignment**, typically manifesting in real-world scenarios, when users question the authenticity of only *specific regions* instead of requesting a *full-image tampering analysis*, they fail to provide context-aware responses.

We first delve into a fundamental question: *Why do existing methods suffer from weak visual-text alignment?* A key underlying factor is **the lack of a guidance mechanism (visual-text alignment at the input-level)** to regulate the MLLMs' visual attention to user-specified image regions. Specifically, as illustrated in Figure 1, when a user asks whether the *fish-shaped kite in the upper right corner* has been manipulated, the MLLM should primarily attend to that region in order to generate a justification regarding its authenticity. However, because existing methods lack a mechanism to adaptively adjust the visual attention, the MLLM still examines the entire image, and then produces static textual explanations, regardless of the query. In addition, during training, they typically only employ generic questions such as "Please find the tampered region(s)," which fail to simulate effective human–machine interaction in real-world scenarios. This design flaw not only causes irrelevant

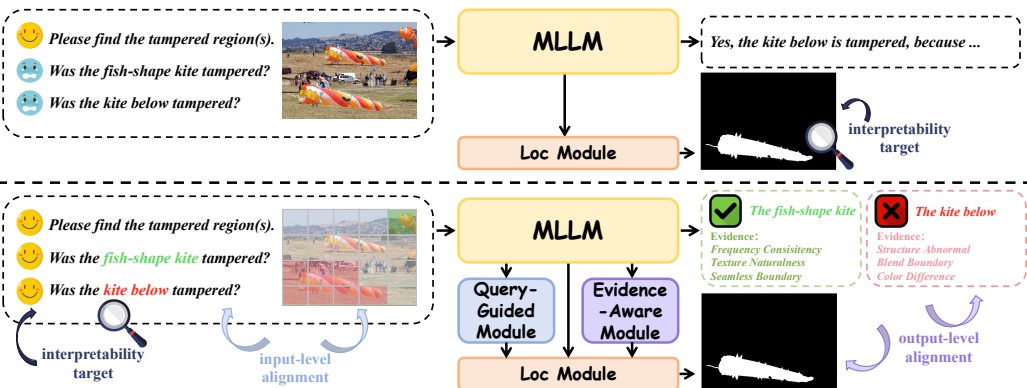

Figure 1: Comparison between existing MLLM-based IFL methods and our approach. Existing methods (upper) often suffer from weak visual-text alignment. In contrast, our approach (lower) addresses this issue by redefining interpretability in IFL as user-query driven and introducing a Query-Guided Module (QGM) and an Evidence-Aware Module (EAM) to enhance visual-text alignment at the input and output levels, respectively.

responses but also leads to suboptimal performance of IFL task. Without an effective guidance mechanism, the MLLM tends to treat every image detail indiscriminately and has to learn manipulation traces from redundant image information through human-provided explanations. While this strategy is feasible, it is highly inefficient. Similarly, another limitation is **the lack of a transfer mechanism (visual-text alignment at the output-level)** to align the useful information embedded in the explanatory text with the visual authenticity evidence required for forgery localization. Although the strong language modeling capability of MLLMs enables them to generate explanations for forensic tasks, existing research largely overlooks aligning the knowledge learned from simpler tasks (explanation/text modality) with more complex tasks (forgery localization/visual modality), which also results in suboptimal IFL performance.

Our analysis has revealed that the weak visual-text alignment of existing MLLM-based IFL methods stems from the absence of a guidance mechanism on the input side and a transfer mechanism on the output side. However, simply enforcing such components without a sound conceptual foundation is infeasible. The deeper obstacle lies in the **global outcome driven** paradigm, which directs interpretability toward forgery localization results and focuses MLLM's visual attention on the entire image. To overcome this limitation, we propose a new paradigm: interpretability in IFL ought to be **regional user-query driven**. In other words, when the user queries are related to the specific regions, the MLLM should regulate its visual attention to those regions and explain the authenticity evidence. To curate the training data to our user-query driven principle, we perform a Question-and-Answer style preprocessing step. In addition to generic queries such as "Please find the tampered region(s)," we construct content-aware questions tailored to each image (e.g., "Was *the kite below* tampered?") together with authenticity evidence. Building on this new principle, we propose **U**ser-query **D**riven **I**mage **S**hield (UDIS), which incorporates two key components. **Query-Guided Module (QGM)** enhances the alignment at the input-level by introducing a [QUERY] token and a visual features filtering process to guide the MLLM focusing on the query-related regions. **Evidence-Aware Module (EAM)** enhances the alignment at the output-level by introducing a [EVI] token and an auxiliary authenticity evidence classification task. Specifically, inspired by (Thakkar et al., 2025), QGM is designed to pre-select the image patch embeddings most relevant to the user's query and aggregate them via the specialized token [QUERY] output by the MLLM. This enables the model to actively focus on image regions corresponding to the query, thereby addressing irrelevant responses and suboptimal IFL performance arising from weak visual-text alignment at the input-level. For EAM, benefiting from the user-query driven principle, we constructed a fine-grained auxiliary authenticity evidence classification task based on the user-query related image region features (output of QGM), so that MLLM can learn the visual representation of the textual authenticity evidence in different image regions and thereby align the explanatory textual knowledge with the visual forgery localization capability more effectively. The modal-consistency knowledge learned through [QUERY] and [EVI] facilitates the specialized [LOC] token output by MLLM in acquiring more precise tamper-

ing features, which is beneficial to improve IFL performance. Extensive experiments demonstrate that the proposed framework provides interpretable, query-focused responses regarding the authenticity of different image regions, highlighting its strong practical value, and achieves state-of-the-art IFL performance. The main contributions in this paper are summarized as follows.

- We introduce a **paradigm shift** by redefining interpretability in IFL as responsive to specific user queries rather than delivering generic explanations. This redefinition establishes a conceptual foundation for overcoming the weak visual-text alignment that constrains existing methods, and provides a new perspective that inspire the development of more advanced MLLM-based IFL frameworks in the future, beyond the specific solution we propose.

- We propose a new framework based on user-query driven paradigm, **U**ser-query **D**riven **I**mage **S**hield (UDIS), which incorporates two key modules—the Query-Guided Module (QGM) and the Evidence-Aware Module (EAM). These modules enhance multimodal alignment in MLLM from two perspectives: aligning user queries with visual attention, and aligning explanatory textual knowledge with forgery localization capability.

- Extensive experiments demonstrate that our framework delivers user-query driven interpretability for specific image regions and achieves superior forgery localization performance compared to existing IFL methods.

## 2 RELATED WORKS

### 2.1 IMAGE FORGERY LOCALIZATION

The IFL task aims to localize tampered regions in images. Traditional methods primarily focus on learning tampering-specific features. DiffForensics (Yu et al., 2024) employs a self-supervised denoising diffusion paradigm to enforce the model to focus on mesoscopic representations. Similarly, Mesorch (Zhu et al., 2025) combines Transformers and CNNs in parallel, where Transformers extract macro-level information and CNNs capture micro-level details, which are then integrated into mesoscopic representations. SparseViT (Su et al., 2025) extracts non-semantic features by reformulating the dense global self-attention in Vision Transformer (ViT) (Dosovitskiy et al., 2020) into a sparse and discrete form. However, these methods lack interpretability and the benefit of rich prior knowledge, and their performance is generally inferior to that of MLLM-based approaches.

### 2.2 MLLMS-BASED IFL METHODS

With the rapid development of MLLMs, leveraging their powerful prior knowledge for IFL tasks has become a research hotspot in recent years. FakeShield (Xu et al., 2024) introduces a multimodal framework that evaluates image authenticity, generates tampered region masks, and provides judgment rationales based on both pixel-level and image-level tampering clues. SIDA (Huang et al., 2025) presents a framework for deepfake detection, localization, and explanation. ForgeryGPT (Liu et al., 2024) advances the IFL task by capturing high-order forensic knowledge correlations of forged images across diverse linguistic feature spaces, while also enabling explainable generation and interactive dialogue through a customized MLLM architecture. HEIE (Yang et al., 2025) introduces the CoT-Driven Explainable Trinity Evaluator, which combines heatmaps, scores, and explanatory outputs, and employs chain-of-thought reasoning to decompose complex tasks into progressively easier subtasks, thereby enhancing interpretability. Despite their promising results, these methods are developed under an outcome-driven paradigm and neglect the importance of multimodal alignment in MLLM-based IFL models, which leads to suboptimal performance and irrelevant responses to user queries.

## 3 METHOD

### 3.1 OVERVIEW

Considering that in real-world scenarios users may question the authenticity of specific regions within an image, it is necessary to account for such targeted queries during training and incorporate structural designs that enhance the MLLM's ability to understand user intent. As illustrated in

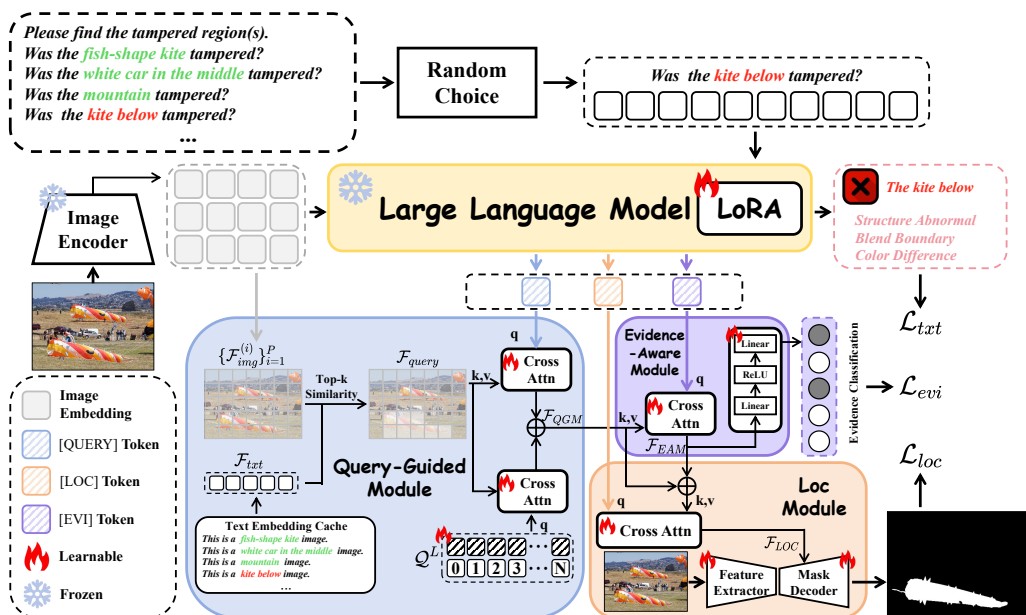

Figure 2: The details of the proposed framework. The Query-Guided Module (QGM) aligns the user queries with the MLLM's visual attention, while the Evidence-Aware Module (EAM) aligns explanatory textual knowledge with forgery localization capability.

Figure 2, the proposed framework consists of four core components: i) an **MLLM** that receives a tampered image and a user query; ii) a **Query-Guided Module (QGM)** that aligns the query with the visual regions of interest attended by the MLLM (enhancing the alignment at the input-level); iii) an **Evidence-Aware Module (EAM)** that aligns the explanatory textual knowledge with forgery localization capability (enhancing the alignment at the output-level); and iv) a **Localization Module** that generates the final prediction.

The MLLM analyzes the relationship between the user query and the image content, and outputs a textual explanation along with three special tokens: a [QUERY] token designed to focus on the image features most relevant to the query; a [EVI] token that learns the visual representations of textual authenticity evidence; and a [LOC] token that encodes tampering traces.

To enhance the visual-text alignment ability of the MLLM at the input-level, we design the Query-Guided Module (QGM). This module introduces a visual feature filtering process and a special token [QUERY]. The core idea of QGM is to filter redundant information from the image embeddings produced by the MLLM's image encoder, guided by text embeddings most relevant to the query. By aggregating the filtered image features through the [QUERY] token, QGM forces the MLLM to focus on specific regions related to user queries. This design alleviates the inefficiency of searching for useful clues within the entire image—a problem that often leads to irrelevant responses and poor forgery localization accuracy. To enhance the visual-text alignment ability of the MLLM at the output-level, we design the Evidence-Aware Module (EAM). This module introduces an auxiliary authenticity evidence classification task and a special token [EVI]. The core idea of EAM is to construct an auxiliary task that takes image region features (fine-grained) as input and predicts authenticity evidence (structured) as output, thereby accommodating the differences in learning objectives between interpretability (structured but coarse) and forgery localization (fine-grained but unstructured). In this way, the MLLM learns visual representations of textual authenticity evidence, and the resulting cross-modal consistency knowledge is stored in the [EVI] token. This design addresses the challenge of transferring explanatory textual knowledge into forgery localization capability. Finally, the [QUERY] and [EVI] tokens, which encode cross-modal consistency knowledge, enhance the tampering features learned by the special [LOC] token produced by the MLLM, thereby improving IFL performance.

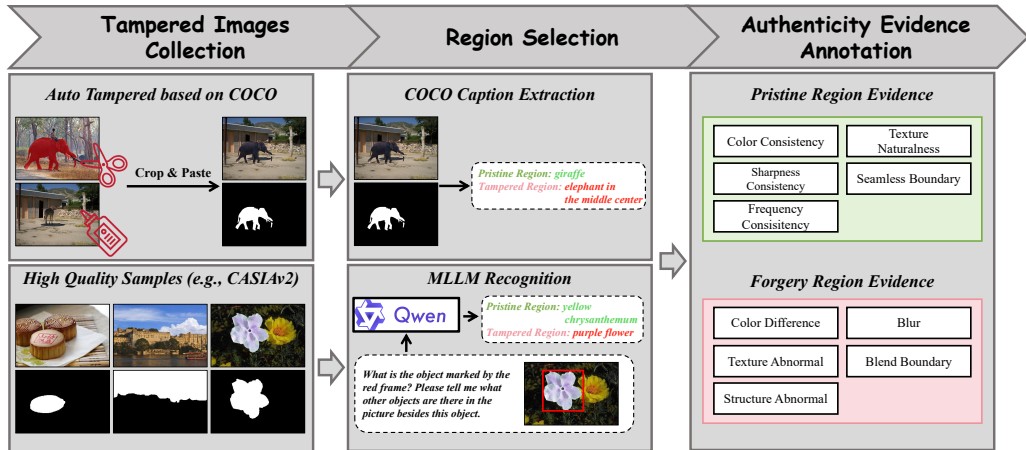

Figure 3: Illustration of training dataset curation, which involves three steps: Tampered Images Collection, Region Selection and Authenticity Evidence Annotation.

## 3.2 DATASET CURATION

To develop a user-query driven IFL model, the training data must include both general questions such as "Please find the tampered region(s)." and specific regional inquiries like "Was the *kite below* tampered?", along with corresponding explanations justifying whether the region is tampered or pristine. As illustrated in Fig. 3, our preprocessing pipeline consists of three steps:

**Tampered Images Collection**. Existing IFL datasets typically lack textual captions for both tampered and pristine regions, and manual annotation is prohibitively labor-intensive. To overcome this, we construct our training set from two complementary sources: i) a limited number of images from public datasets (e.g., CASIAv2 (Dong et al., 2013) and IMD2020 (Novozamsky et al., 2020)), which provide realistic tampered content but still require textual annotations; ii) a large number of auto tampered images based on COCO (Lin et al., 2014), which offers object-level captions.

**Region Selection**. We directly extract the object-level captions that COCO-based tampered images already come with as annotations. For public datasets, we employ Qwen2.5-VL-7B-Instruct (Bai et al., 2023) to recognize image contents. These captions are used as the keywords of the queries.

**Authenticity Evidence Annotation**. Following the forgery type decision method in (Sun et al., 2025), we distinguish potential traces of tampered regions into five categories: Color Difference, Blur, Texture Abnormal, Structure Abnormal, and Blend Boundary. Similarly, for pristine regions, we utilize decision criteria based on natural consistency: Color Consistency, Texture Naturalness, Sharpness Consistency, Seamless Boundary and Frequency Consistency. It is worth noting that multiple types of evidence may co-occur within the same tampered or pristine region, thus this step is a multi-class annotation.

The details of dataset curation process are described in Appendix A.2

## 3.3 QUERY-GUIDED MODULE

In this section, we elaborate on the detailed design of the Query-Guided Module (QGM). Considering that the image features extracted by the MLLM's image encoder primarily capture global semantic information (as shown in Section 4.6), and that the MLLM's original visual-text alignment capability at the input-level may be weakened during IFL training, it becomes necessary to introduce an additional module to reinforce the MLLM's multimodal alignment ability. This enables the model to more effectively comprehend the image according to the user's query and thereby answer diverse questions. Moreover, since tampered regions typically occupy only a small portion of the image, relying on the global features extracted by MLLM's image encoder and its inherent multimodal alignment capability between the input image and textual annotation to detect subtle tampering traces is inefficient, leading to suboptimal forgery localization performance.

To address this, we construct **a visual feature filtering process** anchored on the user query: identifying the image features most relevant to the query and allowing the MLLM to actively learn from these filtered features. However, directly feeding these filtered features into the MLLM may compromise the completeness of the image input. In particular, when the user query is unrelated to the tampered area, the MLLM may fail to accomplish the IFL task. Therefore, while preserving the complete image information input, we employ the [QUERY] token output by MLLM to actively aggregate the filtered image features. Importantly, although user queries may sometimes refer to pristine regions, learning from non-tampered examples can also enhance the MLLM's understanding of the IFL task.

Specifically, the QGM takes the image features output by the MLLM's image encoder and produces filtered image features that are most relevant to the user's query. Most existing MLLM image encoders are implemented based on the Vision Transformer (ViT) (Dosovitskiy et al., 2020) architecture, which divides the image into multiple patches, each represented by a patch embedding $\{\mathcal{F}_{img}^{(i)}\}_{i=1}^{P}$, where $P$ is the number of patches. Our filtering process selects the patch embeddings $\mathcal{F}_{query}$ as follows:

$$\mathcal{F}_{query} = \{\mathcal{F}_{img}^{(i)} \mid i \in \text{Top-}k\big(\{\mathcal{F}_{img}^{(i)} \cdot \mathcal{F}_{txt}^{\top}\}_{i=1}^{P}\big)\}, \tag{1}$$

where $\mathcal{F}_{txt}$ is the text embedding containing the keywords from the user query (e.g., "Was the *elephant* tampered?" and "This is an *elephant* image."), and Top-$k(\cdot)$ refers to the procedure of retrieving the indices corresponding to the $k$ highest-ranked values within a given set. After selecting the target patches, the [QUERY] token output by the MLLM actively aggregates $\mathcal{F}_{query}$ through a Cross-Attention (CA) layer. Given that the IFL task often requires fine-grained features, and the [QUERY] token alone has limited capacity to encode spatially diverse information, we introduce $N$ learnable query vectors $\mathcal{Q}^{L}$, with positional embeddings in the QGM. These $N$ vectors also interact with $\mathcal{F}_{query}$, and the two feature sets are then summed to obtain the output features $\mathcal{F}_{QGM}$:

$$\mathcal{F}_{QGM} = CA(\mathcal{T}_{[\text{QUERY}]}, \mathcal{F}_{query}) + CA(\mathcal{Q}^{L}, \mathcal{F}_{query}), \tag{2}$$

where $\mathcal{T}_{[\text{QUERY}]}$ represents the [QUERY] token, and $CA(\cdot, \cdot)$ denotes the Cross-Attention operation. Notably, during training phase, if the user's query does not refer to any specific region, we simply set $\{\mathcal{F}_{img}^{(i)}\}_{i=1}^{P}$ as $\mathcal{F}_{query}$. In test time, the visual feature filtering process is not performed.

## 3.4 EVIDENCE-AWARE MODULE

In this section, we elaborate on the detailed design of the Evidence-Aware Module (EAM). Given the strong language learning capability of existing MLLM, it is relatively easy for them to acquire explanatory forensic texts. However, a major challenge arises when transferring forensic knowledge learned from text-based tasks to IFL task: the inherent discrepancy between the supervisory signals of the two modalities. Specifically, text annotations provide structured knowledge but are coarse-grained, while pixel-level tampered masks offer fine-grained authenticity information but lack consistent patterns across different samples (detailed illustration is in Appendix A.3). When both types of supervision are imposed directly on the [LOC] token, the MLLM struggles to learn coherent knowledge.

To enable the MLLM to learn **modality-consistenty knowledge**, we first seek a type of information that is jointly required by both explanatory and forgery localization tasks and use it to define an auxiliary training objective. We focus on **authenticity evidence** because it is explicitly grounded in user-specified regions within explanatory texts and directly corresponds to the forensic cues that IFL relies on. Treating authenticity evidence as the auxiliary supervisory signal therefore provides a natural bridge between the textual explanations and forgery localization.

Building on this insight, we formulate a **fine-grained auxiliary authenticity evidence classification task** for user-specific regions. Since localization requires fine-grained representations, the classifier operates on the region-specific features $\mathcal{F}_{QGM}$. By incorporating these region features with supervision of authenticity evidence categories derived from textual annotations, the auxiliary task forces the MLLM to map subtle visual patterns to the structured knowledge used in explanatory annotations. In this way, explanatory textual knowledge is effectively transferred into forgery localization capability, improving cross-modal alignment.

Formally, we take the [EVI] token output by the MLLM, $\mathcal{T}_{[\text{EVI}]}$, as the query, and $\mathcal{F}_{QGM}$ as key and value, which are fed into a CA layer. The resulting feature $\mathcal{F}_{EAM}$ is then passed to a lightweight

classification head $H_{evi}$ to produce the classification result $\hat{E}$:

$$\mathcal{F}_{EAM} = CA(\mathcal{T}_{\texttt{[EVI]}}, \mathcal{F}_{QGM}),$$
$$\hat{E} = H_{evi}(\mathcal{F}_{EAM}). \tag{3}$$

Compared to FakeShield (Xu et al., 2024) which attempts to align explanatory textual knowledge with the IFL capability employing a large language model, our EAM offers two key advantages: i) we explicitly identify authenticity evidence—knowledge that is inherently shared across modalities—as the alignment target, thereby achieving more efficient knowledge transfer; and ii) the fine-grained auxiliary task built on QGM's output features is more compatible with the IFL objective. Moreover, the lightweight design of EAM significantly reduces computational overhead.

### 3.5 LOCALIZATION MODULE

In this section, we provide a detailed description of the Localization Module and the loss functions. We adopt ConvNeXt (Liu et al., 2022) as the feature extractor for the Localization Module. A CA layer is employed to fuse the [LOC] token $\mathcal{T}_{\texttt{[LOC]}}$ generated by the MLLM with $\mathcal{F}_{QGM}$ and $\mathcal{F}_{EAM}$. The fused features are then fed into the Mask Decoder of SAM (Kirillov et al., 2023), denoted as $H_{loc}$, to obtain the predicted tampering mask $\hat{M}$. This process can be formulated as:

$$\mathcal{F}_{LOC} = CA(\mathcal{T}_{\texttt{[LOC]}}, \mathcal{F}_{QGM} + \mathcal{F}_{EAM}),$$
$$\hat{M} = H_{loc}(\mathcal{F}_{LOC}, \mathcal{F}_{conv}), \tag{4}$$

where $\mathcal{F}_{conv}$ denotes the image features extracted by ConvNeXt.

The overall training objective is defined as:

$$\mathcal{L} = \mathcal{L}_{loc}(\hat{M}, M) + \mathcal{L}_{evi}(\hat{E}, E) + \lambda_{txt}\mathcal{L}_{txt}(\hat{T}, T), \tag{5}$$

where $\mathcal{L}_{loc}$ is the loss for IFL, $\mathcal{L}_{evi}$ corresponds to the auxiliary classification loss in EAM, and $\mathcal{L}_{txt}$ is the text prediction loss. Here, $M$ and $E$ denote the pixel-level tampered mask and the authenticity evidence classification label, respectively, while $\hat{T}$ and $T$ represent the MLLM-generated text and the ground-truth text. Both $\mathcal{L}_{evi}$ and $\mathcal{L}_{txt}$ are Cross-Entropy (CE) loss, whereas $\mathcal{L}_{loc}$ is formulated as a combination of Dice loss and BCE loss. A balancing weight $\lambda_{loc}$ is further introduced to regulate the model's attention between tampered and authentic regions:

$$\mathcal{L}_{loc} = \mathcal{L}_{dice}(\hat{M}, M) + \lambda_{loc}\mathcal{L}_{bce}(\hat{M}, M). \tag{6}$$

## 4 EXPERIMENT

### 4.1 EXPERIMENTAL SETUP

**Dataset**: We constructed the training dataset as described in Section 3.2. It consists of 20k auto tampered samples based on COCO (Lin et al., 2014) and high-quality tampered images collected from CASIAv2 (Dong et al., 2013) and IMD2020 (Novozamsky et al., 2020). For evaluation, we used CASIAv1 (Dong et al., 2013), NIST16 (Guan et al., 2019), DSO-1 (De Carvalho et al., 2013), Korus (Korus & Huang, 2016), AutoSplice (Jia et al., 2023), and CocoGlide (Guillaro et al., 2023) (more details are in Appendix A.2).

**Evaluation Criteria**: For localization performance, we report F1 score and Intersection over Union (IoU), using a consistent image size of 512×512 and fixed threshold of 0.5 during computation. We also use the Area Under the Receiver Operating Characteristic Curve (AUC) score, computed without a threshold, to assess the performance of different IFL methods. **It is worth noting that**, for other MLLM-based methods and our proposed approach, we adopted the standardized input query "Please find the tampered region(s)." during testing to ensure fairness.

**Implementation Details**: We used LISA-7B (Lai et al., 2024) as the pre-trained model to leverage its reasoning and segmentation capabilities. The MLLM was fine-tuned via LoRA ($\alpha$=16, dropout rate=0.05), with input images resized to 512×512. The initial learning rate was set to $3 \times 10^{-4}$ with a batch size of 4 and gradient accumulation steps of 10. $\lambda_{txt}$ and $\lambda_{loc}$ were set to 0.5. The Text Embedding Cache of QGM was initialized by CLIP's text encoder (Radford et al., 2021).

Table 1: IFL performance comparison. * indicates the pre-trained weights provided by the original study. The best and second-best results are highlighted in **bold** and underline, respectively.

| Methods | CASIAv1 | | NIST16 | | DSO-1 | | Korus | | AutoSplice | | CocoGlide | | Average | |
|---|---|---|---|---|---|---|---|---|---|---|---|---|---|---|
| | F1 | IoU | F1 | IoU | F1 | IoU | F1 | IoU | F1 | IoU | F1 | IoU | F1 | IoU |
| MVSS-Net* | .539 | .470 | .292 | .239 | .271 | .188 | .095 | .067 | .333 | .241 | .356 | .275 | .314 | .247 |
| IML-ViT | .600 | .537 | .383 | .306 | .286 | .185 | .215 | .158 | .299 | .201 | .394 | .311 | .363 | .283 |
| APSC-Net | .328 | .286 | .317 | .275 | .187 | .150 | .175 | .137 | .366 | .281 | .102 | .078 | .246 | .201 |
| SparseViT | .614 | .560 | .299 | .252 | .157 | .118 | .093 | .066 | .340 | .248 | .409 | .341 | .319 | .264 |
| Mesorch | .560 | .524 | .181 | .153 | .045 | .031 | .057 | .041 | .245 | .170 | .266 | .217 | .226 | .189 |
| SIDA* | .331 | .274 | .295 | .241 | .084 | .061 | .166 | .118 | .479 | .355 | .412 | .325 | .295 | .229 |
| FakeShield* | .584 | .530 | .263 | .232 | .436 | **.398** | .098 | .077 | .379 | .312 | .547 | .449 | .385 | .333 |
| UDIS(ours) | **.665** | **.593** | **.454** | **.371** | **.529** | **.398** | **.352** | **.263** | **.638** | **.503** | **.644** | **.535** | **.547** | **.444** |

Table 2: Comparison of interpretability.

| Methods | $HR_{query}$ | $Acc_{r/f}$ |
|---|---|---|
| SIDA* | .636 | .211 |
| FakeShield* | .305 | .523 |
| UDIS w/o QA,QGM | .214 | .584 |
| UDIS w/o QGM | .896 | .700 |
| **UDIS(ours)** | **.912** | **.732** |

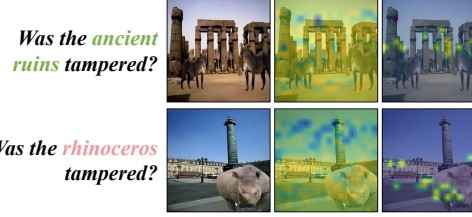

Figure 4: Visualization of image features.

## 4.2 COMPARISON OF IFL

To validate the effectiveness of our proposed method, we compared it with multiple IFL models, including MVSS-Net (Chen et al., 2021), IML-ViT (Ma et al., 2023), APSC-Net (Qu et al., 2024), SparseViT (Su et al., 2025), Mesorch (Zhu et al., 2025), SIDA (Huang et al., 2025), and FakeShield (Xu et al., 2024). For methods with publicly available training code, such as IML-ViT (Ma et al., 2023), APSC-Net (Qu et al., 2024), SparseViT (Su et al., 2025), and Mesorch (Zhu et al., 2025), we retrained the models using the same training dataset to ensure fairness. The evaluation results are summarized in Table 1, which show that our method achieves state-of-the-art performance across multiple datasets. In addition, Appendix A.4.1 reports the pixel-level AUC scores, as well as detailed analyses of localization and explanatory outputs.

## 4.3 COMPARISON OF INTERPRETABILITY

To evaluate the ability of UDIS to perceive the key regions mentioned in the user query—i.e., whether it can answer according to the query—we design the metric $HR_{query}$ (Hit Rate),

$$HR_{query} = \frac{A_{hit}}{T}, \tag{7}$$

where $T$ is the number of test samples and $A_{hit}$ is the number of the answer containing the keyword from the user query (e.g., "Was the *elephant* tampered" and "The *elephant* is fake, because ..."). In addition, we assess the accuracy of different methods to judge the authenticity of the regions specified by the user, denoted as $Acc_{r/f}$. We employ CAISAv1 and AutoSplice to perform the experiment, and report the average metric in Table 2. It indicates that UDIS has a stronger ability to provide targeted answers to user questions. Moreover, the Question-and-Answer style preprocessing (denoted as "QA" in Table 2 and Table 3) and QGM are also key factors to enhance this capability.

## 4.4 ROBUSTNESS EVALUATION

To evaluate the robustness of our method against post-processing attacks in real-world scenarios, we applied various post-processing operations, including JPEG Compression, Gaussian Blur, Gaussian Noise, and Median Blur. As shown in Figure 5 (tested on CASIAv1, NIST16, DSO-1 and Korus),

Table 3: Ablation study for different components. "QA" indicates that, in addition to "Please find the tampered region(s).", the training data also includes questions involving specific regions.

| | Components | | | Average | | |
|---|---|---|---|---|---|---|
| | QA | QGM | EAM | F1 | IoU | AUC |
| (1) | - | - | - | .455 | .349 | .825 |
| (2) | - | ✓ | ✓ | .461 | .368 | .842 |
| (3) | ✓ | - | - | .462 | .363 | .822 |
| (4) | ✓ | ✓ | - | .500 | .401 | .839 |
| (5) | ✓ | - | ✓ | .489 | .395 | .855 |
| (6) | ✓ | ✓ | ✓ | **.547** | **.444** | **.856** |

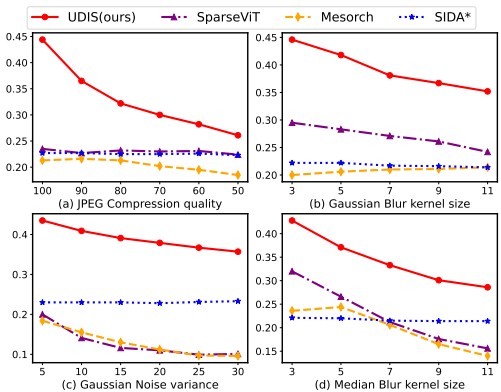

Figure 5: Robustness evaluation against four post-processing methods.

our method achieves superior performance under different types and intensities of post-processing, demonstrating stronger robustness. It is worth noting that our method was not trained with any post-processing attack augmentations. The robustness evaluation of interpretability is in Appendix A.4.2.

## 4.5 ABLATION STUDY

To validate the core design principle we proposed for MLLM-based IFL methods—user-query driven—and the effectiveness of the corresponding components (QGM and EAM), we evaluated the frogery localization capability of the model under different experimental settings. The ablation studies were conducted on all test datasets and the averaged metrics were reported. As shown in Table 3, a comparison between (1) and (2), as well as between (1) and (3), indicates that individually applying either the architectural design or Question-and-Answer style changes yields no significant improvement in IFL capability. However, when the approach is designed according to the user-query driven principle (both architecture and QA style), the improvement observed in the comparison between (1) and (6) becomes substantial (F1 increased by 9.2%, IoU increased by 9.5%). Furthermore, a comparison between (3) and (4), as well as between (3) and (5), shows that QGM and EAM, by enhancing the alignment at the input and output level respectively, further strengthen the forgery localization capability of the model (F1 improves by 3.8% and 2.7%, respectively). We also evaluate the influence of the Top-$k$ parameter of QGM in Appendix A.4.3 and the comparison of IFL performance between UDIS and just employing Localization Module in Appendix A.4.4.

## 4.6 VISUALIZATION OF IMAGE FEATURES

In Figure 4, we visualize $\{\mathcal{F}_{img}^{(i)}\}_{i=1}^{P}$ (second column) and $\mathcal{F}_{query}$ (third column). It can be observed that the selected patch embeddings of QGM are indeed related to the user query, thereby focusing the model's attention on user-specific regions and improving both the accuracy of the responses and the IFL capability. More details are illustrated in Appendix A.4.5.

## 5 CONCLUSION

In this paper, we addressed weak visual-text alignment in MLLM-based IFL methods. We introduced a new principle—interpretability should be regional user query driven—and proposed the UDIS framework to realize it. With the Query-Guided Module (QGM) and Evidence-Aware Module (EAM), UDIS improves multimodal alignment from both input and output perspectives, enabling context-aware responses and accurate localization. Experiments show that UDIS achieves state-of-the-art performance while offering stronger interpretability and practical utility. In the future, we will further explore the multi-round dialogue capabilities of UDIS and try to use more flexible visual feature filtering methods in QGM.

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

# A APPENDIX

## A.1 THE USE OF LARGE LANGUAGE MODELS (LLMS)

In the writing of this paper, we used large language models (LLMs) to check grammar and spelling correctness, as well as for minor language polishing.

## A.2 MORE DETAILS OF DATASETS

As described in Section 3.2, to implement our proposed user-query driven principle, we supplement and annotate existing datasets. The detailed procedure is as follows:

**Tampered Images Collection**: For auto tampered samples, we use COCO (Lin et al., 2014) as the source dataset. The auto tampered samples cover two common forgery types: copy-move and splicing. Concretely, an object is randomly cropped from image $I_1 \in \mathbb{R}^{H_1 \times W_1 \times 3}$, resized with a random scaling factor, and then pasted either back into $I_1$ (copy-move) or into another image $I_2 \in \mathbb{R}^{H_2 \times W_2 \times 3}$ (splicing). Following this procedure, we construct a dataset containing 20k manipulated images, with 10k images for copy-move and 10k for splicing.

**Region Selection**: To simulate real-world scenarios in which users may pose queries about specific regions in an image, we provide object-level annotations for each sample. For auto tampered images, we directly use the object labels from COCO (Lin et al., 2014). In cases where identical objects appear (e.g., in copy-move forgeries), we disambiguate them by adding positional information (e.g., "the elephant in the middle center"). For public datasets lacking object-level annotations (e.g., CASIAv2 (Dong et al., 2013) and IMD2020 (Novozamsky et al., 2020)), we employ Qwen2.5-VL-7B-Instruct (Bai et al., 2023). Specifically, the tampered region is marked with a red bounding box, and Qwen is prompted to recognize both the content inside the box and other objects outside it. All automatically generated captions are subsequently verified manually for correctness.

**Authenticity Evidence Annotation**: In addition to authenticity evidence for tampered regions, we also define multiple types of evidence for pristine regions, which are summarized as follows:

i) Color Consistency: compute mean and variance differences in Lab color space between the region and the remaining.

ii) Texture Naturalness: assess local entropy (via GLCM or rank entropy) between the region and its surroundings.

iii) Sharpness Consistency: apply the Laplacian operator and compare variance values.

iv) Seamless Boundary: use Sobel gradients and Canny edges to evaluate boundary smoothness.

v) Frequency Consistency: perform a 2D DCT and compare high-/low-frequency energy ratios.

These five criteria quantify the natural consistency of pristine regions and provide reliable evidence of authenticity. We use a fixed threshold to determine whether these evidences are valid. If the current sample does not meet the criteria of any evidence, the loss value of this part of the sample is directly set to zero when calculating $\mathcal{L}_{evi}$. Details of the training and testing datasets, as well as illustrative examples, are provided in Table 4 and Figure 6. Notably, we only employ tampered images for training and testing.

## A.3 MORE ILLUSTRATIONS OF EAM'S MOTIVATION

To more accurately describe the discrepancy between pixel-level tampered masks and text annotations discussed in Section 3.4, Figure 7 illustrates the following: although most of the information in tampered images remains the same (e.g., pasting the same object onto an identical image in different ways), the pixel-level tampered masks can take completely different forms (in terms of location, size, and shape of the tampered regions). In contrast, the authenticity evidence in the interpretive annotations remains consistent. This highlights that pixel-level tampered masks lack consistent patterns across samples, whereas text annotations generally exhibit more structured knowledge. From another perspective, this discrepancy also reflects that pixel-level tampered masks are more fine-grained, while text annotations are relatively coarse-grained.

Table 4: Details of the training and testing datasets. Com., Spl., and Inp. denote copy-move, splicing, and inpainting.

| Datasets | Pristine | Tampered | Com. | Spl. | Inp. |
|---|---|---|---|---|---|
| *#Training* | | | | | |
| COCO-based auto tampered | 0 | 20000 | ✓ | ✓ | - |
| CASIAv2 | 7491 | 5036 | ✓ | ✓ | - |
| IMD2020 | 414 | 2010 | - | ✓ | - |
| *#Testing* | | | | | |
| CASIAv1 | 800 | 920 | ✓ | ✓ | - |
| NIST16 | 564 | 560 | ✓ | ✓ | ✓ |
| DSO-1 | 0 | 100 | - | ✓ | - |
| Korus | 0 | 220 | - | ✓ | - |
| AutoSplice | 2273 | 3621 | - | - | ✓ |
| CocoGlide | 512 | 512 | - | - | ✓ |

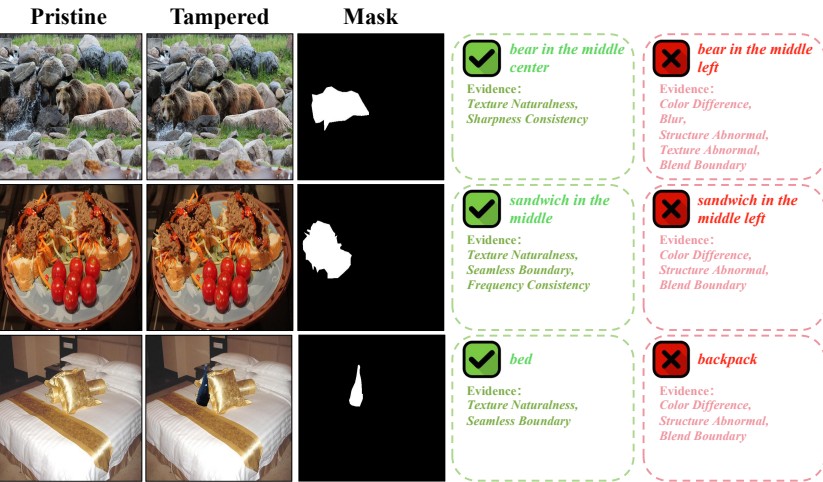

Figure 6: Examples of our training dataset. The interpretability annotations are formatted as "{authenticity},{region},{authenticity evidence}" (e.g., real,bear in the middle center,texture naturalness,sharpness consistency)

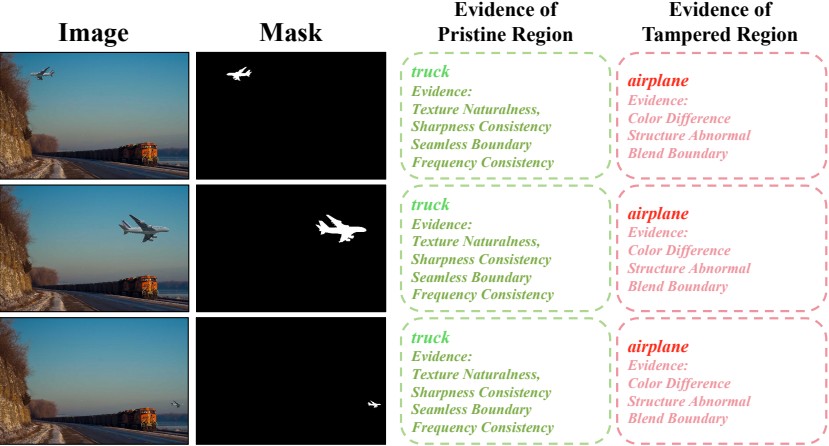

Figure 7: The illustration of EAM's motivation. The same object (an airplane) is randomly pasted at different positions and scales within an identical image, and annotated using the evidence decision method described in A.2.

Table 5: Pixel-level AUC score comparison of different IFL methods. * indicates the pre-trained weights provided by the original study. The best and second-best results are highlighted in **bold** and underline, respectively.

| Methods | CASIAv1 | NIST16 | DSO-1 | Korus | AutoSplice | CocoGlide | Average |
|---------|---------|--------|-------|-------|------------|-----------|---------|
| MVSS-Net* | .887 | .792 | .732 | .641 | .839 | .824 | .786 |
| IML-ViT | .904 | .839 | .823 | .705 | .885 | .876 | .839 |
| APSC-Net | .652 | .646 | .577 | .570 | .644 | .542 | .605 |
| SparseViT | .911 | **.856** | .808 | .727 | **.925** | .889 | .853 |
| Mesorch | .911 | .812 | .668 | .651 | .796 | .828 | .778 |
| SIDA* | .789 | .771 | .645 | .671 | .862 | .803 | .757 |
| FakeShield* | .859 | .676 | .756 | .556 | .706 | .835 | .731 |
| UDIS(ours) | **.913** | .835 | **.856** | **.747** | .886 | **.896** | **.856** |

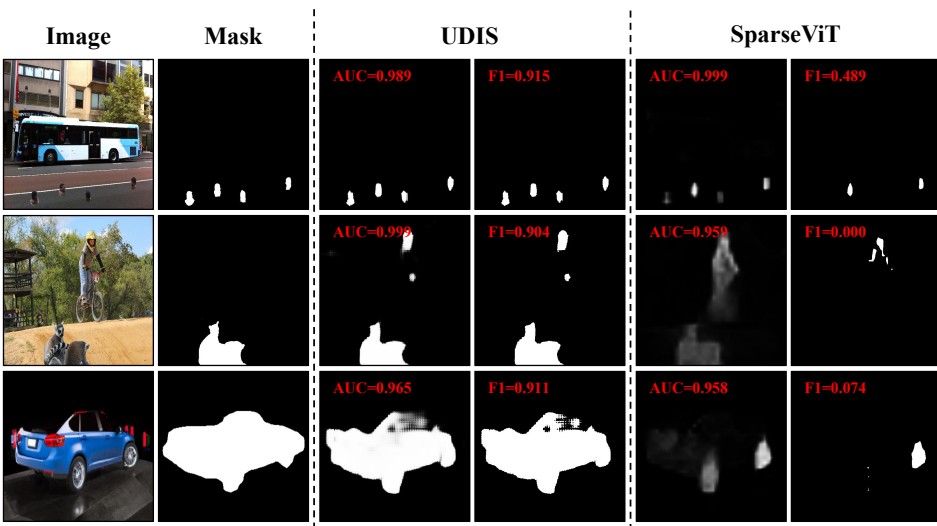

Figure 8: Comparison of the outputs of different models and the forgery localization results after thresholding.

Table 6: Robust evaluation of interpretability. Tested on CASIAv1

| Post Processing | SIDA | | FakeShield | | UDIS | |
|-----------------|------|------|------------|------|------|------|
| | $HR_{query}$ | $Acc_{r/f}$ | $HR_{query}$ | $Acc_{r/f}$ | $HR_{query}$ | $Acc_{r/f}$ |
| - | .636 | .211 | .290 | .524 | .905 | .722 |
| JPEG 100 | .623 | .274 | .298 | .401 | .888 | .767 |
| JPEG 50 | .604 | .272 | .306 | .411 | .885 | .750 |
| Gaussian Blur 3 | .640 | .262 | .292 | .399 | .887 | .749 |
| Gaussian Blur 11 | .612 | .264 | .286 | .451 | .891 | .742 |
| Gaussian Noise 5 | .626 | .273 | .284 | .443 | .890 | .742 |
| Gaussian Noise 30 | .618 | .264 | .278 | .408 | .891 | .739 |
| Median Blur 3 | .615 | .275 | .293 | .508 | .891 | .740 |
| Median Blur 11 | .623 | .297 | .292 | .410 | .892 | .736 |

Table 7: Influence of the Top-$k$ parameter of QGM.

| Top-$k$ | Average | | |
|---|---|---|---|
| | F1 | IoU | AUC |
| 4 | .472 | .371 | .824 |
| 16 | .547 | .444 | .856 |
| 32 | .524 | .425 | .844 |
| 256 | .528 | .429 | .849 |

Table 8: Verifying the importance of the MLLM in IFL task.

| | Average | | |
|---|---|---|---|
| | F1 | IoU | AUC |
| Loc Module | .522 | .426 | .847 |
| UDIS(ours) | **.547** | **.444** | **.856** |

## A.4 MORE EXPERIMENT RESULTS

### A.4.1 MORE COMPARISON EXPERIMENTS

In addition to F1 and IoU metric, we also evaluated the pixel-level AUC score of different IFL methods in Table 5. The evaluation results demonstrate that UDIS achieves the best performance compared to other state-of-the-art methods.

From the experimental results, it can be observed that the pixel-level AUC scores of MLLM-based IFL methods are generally lower. For example, UDIS does not achieve the best AUC performance on NIST16 and AutoSplice. Compared with feature-extraction-based methods such as SparseViT (Su et al., 2025), the improvement in AUC is also less significant than that in F1 and IoU, which are threshold-dependent metrics. To further investigate this, we analyzed the differences between the outputs of MLLM-based IFL methods and those of traditional feature-extraction-based methods. As illustrated in Figure 8, the raw output of UDIS (before thresholding) shows a clear distinction between tampered and authentic regions: the logits in tampered regions are very close to 1, while those in authentic regions are very close to 0. This phenomenon may be related to the SAM Decoder, and has also been observed in methods such as SIDA (Huang et al., 2025) and FakeShield (Xu et al., 2024). In contrast, the outputs of SparseViT (Su et al., 2025) rarely approach 1, even in tampered regions. Consequently, when computing the threshold-independent AUC score, SparseViT (Su et al., 2025) can achieve better performance under conditions involving smaller thresholds. However, when using fixed thresholds to compute metrics such as F1 or IoU, methods like SparseViT (Su et al., 2025) may fail to identify tampered regions due to their relatively low output values. Considering that, in practical scenarios, tampering localization is usually performed with fixed thresholds, models with higher F1 and IoU scores are generally more aligned with real-world application requirements. The forgery localization and interpretablity results are shown in Figure 11, 12, 13 and 14.

### A.4.2 MORE ROBUSTNESS EVALUATION

In Table 6, we evaluate the impact of different image post-processing techniques on the interpretability of MLLM-based IFL methods. The results show that UDIS is able to maintain strong interpretability across various post-processing conditions.

### A.4.3 INFLUENCE OF TOP-$k$ OF QGM

To examine the impact of the Top-$k$ parameter in the visual feature filtering process of QGM on forgery localization performance, we evaluated the performance of UDIS under different parameter settings. As shown in Table 7, the results demonstrate that when the Top-$k$ parameter is sufficiently large (greater than or equal to 16), satisfactory localization performance can be achieved.

### A.4.4 IMPORTANCE OF THE MLLM

To evaluate the importance of the MLLM in UDIS for the IFL task, as shown in Table 8, we compare the forgery localization performance of using only the Loc Module (ConvNeXt + SAM's Decoder, where $\mathcal{F}_{LOC}$ is replaced with a learnable vector) against the full UDIS framework. The results demonstrate that UDIS achieves overall better localization capability. **More importantly**, the introduction of the MLLM further provides interpretability, an essential ability for human–machine interaction in real-world scenarios.

Figure 9: Visualization results of $\mathcal{F}_{query}$ under different problems.

**Templates without specific regions**

(1) Please find "the digitally manipulated part(s)".

(2) Please find "the altered region(s)".

(3) Please find "the edited area(s)".

(4) Please find "the forged region(s)" .

(5) Please find "the tampered part(s)".

(6) Please find "the modified segment(s)".

(7) Please find "the fake content(s)".

(8) Please find "the inconsistent area(s) in the image".

(9) Please find "the visually tampered region(s)".

(10) Please find "the abnormal pattern(s) indicating manipulation".

**Templates with specific regions**

(1) Has the {Region} been artificially introduced into the original image?

(2) Is the {Region} a manipulated or edited element within this image?

(3) Was the {Region} digitally inserted into the image?

(4) Does the {Region} appear to be a product of image editing?

(5) Was the {Region} part of the original scene, or added later via modification?

(6) Is there any indication that the {Region} was superimposed onto the image?

(7) Could the {Region} be an artificial addition rather than a naturally occurring element in the image?

(8) Is the presence of the {Region} consistent with the original, unaltered content of the image?

(9) Does visual evidence suggest that the {Region} has been digitally altered or introduced?

(10) Was the {Region} included through post-processing rather than captured in the original shot?

Figure 10: The design details of question templates used for training.

### A.4.5 VISUALIZATION OF $\mathcal{F}_{query}$

We asked questions about different regions of the same image and visualized the image regions selected by QGM. The results in Figure 9 show that the image regions selected by QGM do indeed address the user's needs.

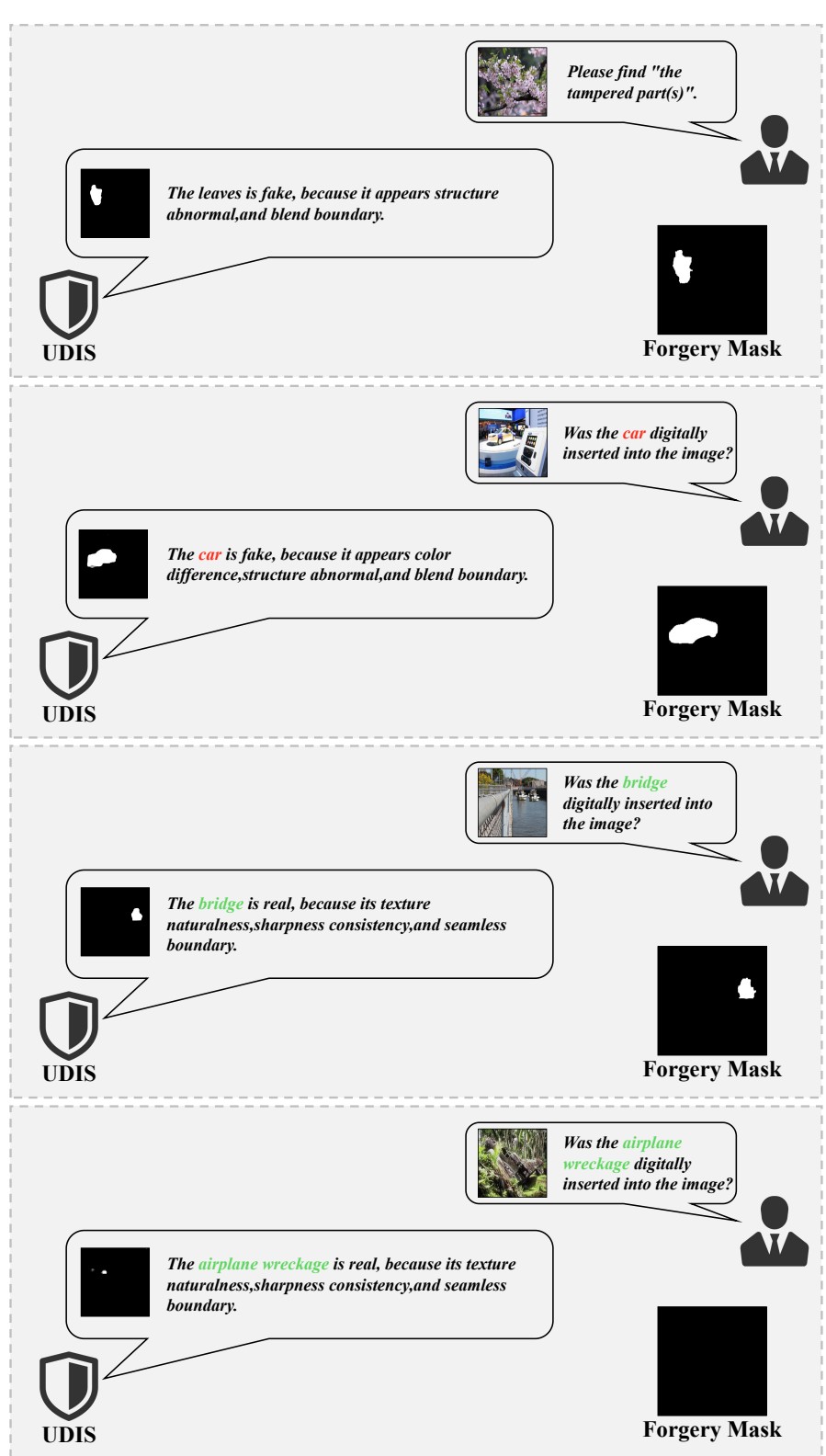

Figure 11: Examples of UDIS's forgery localization and interpretability results. For convenient human–machine interaction, we format the interpretive output of UDIS as: "The {region} is {authenticity}, because its/it appears {authenticity evidence}."

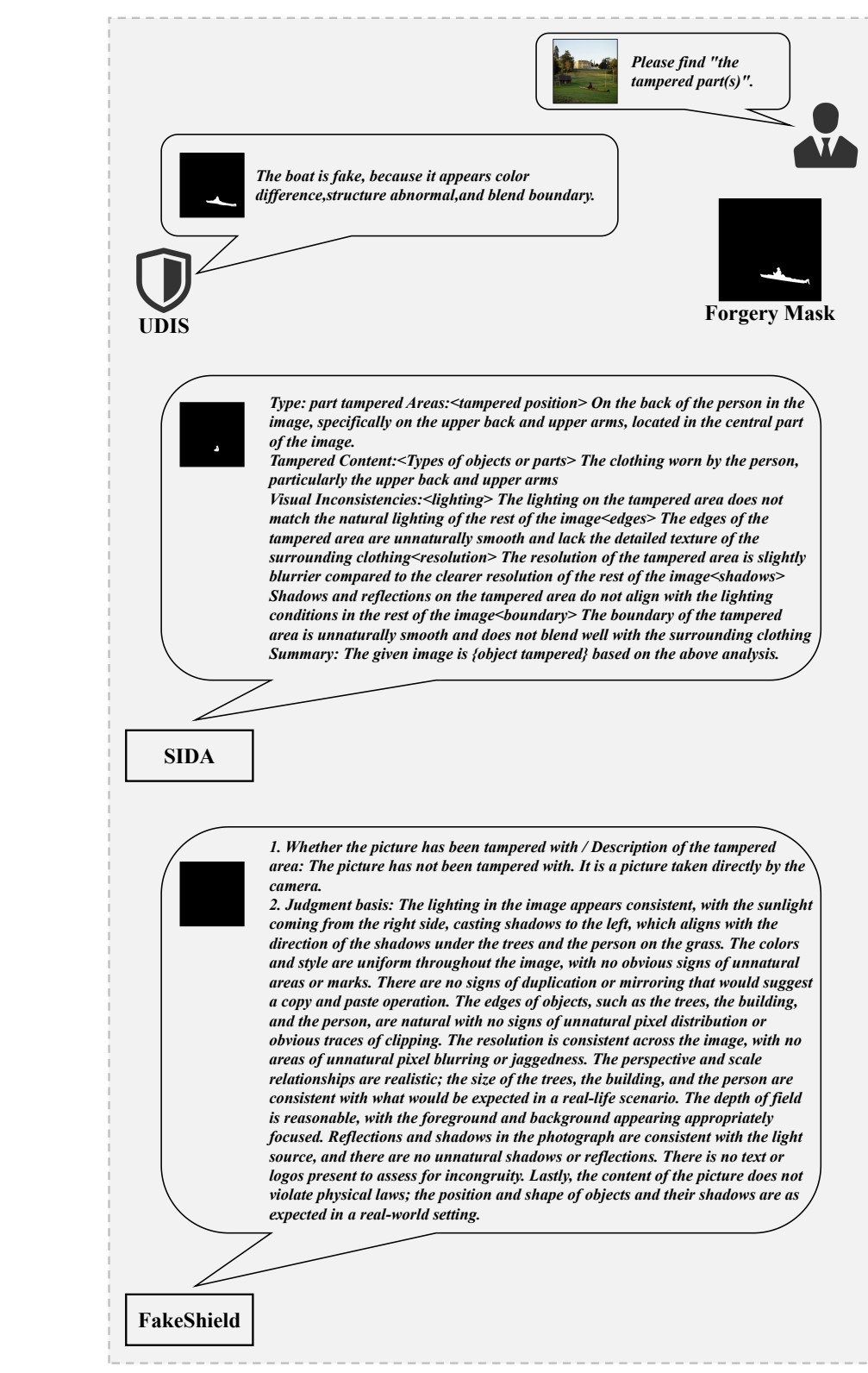

Figure 12: Comparison of forgery localization and interpretability results among different MLLM-based methods. (common query)

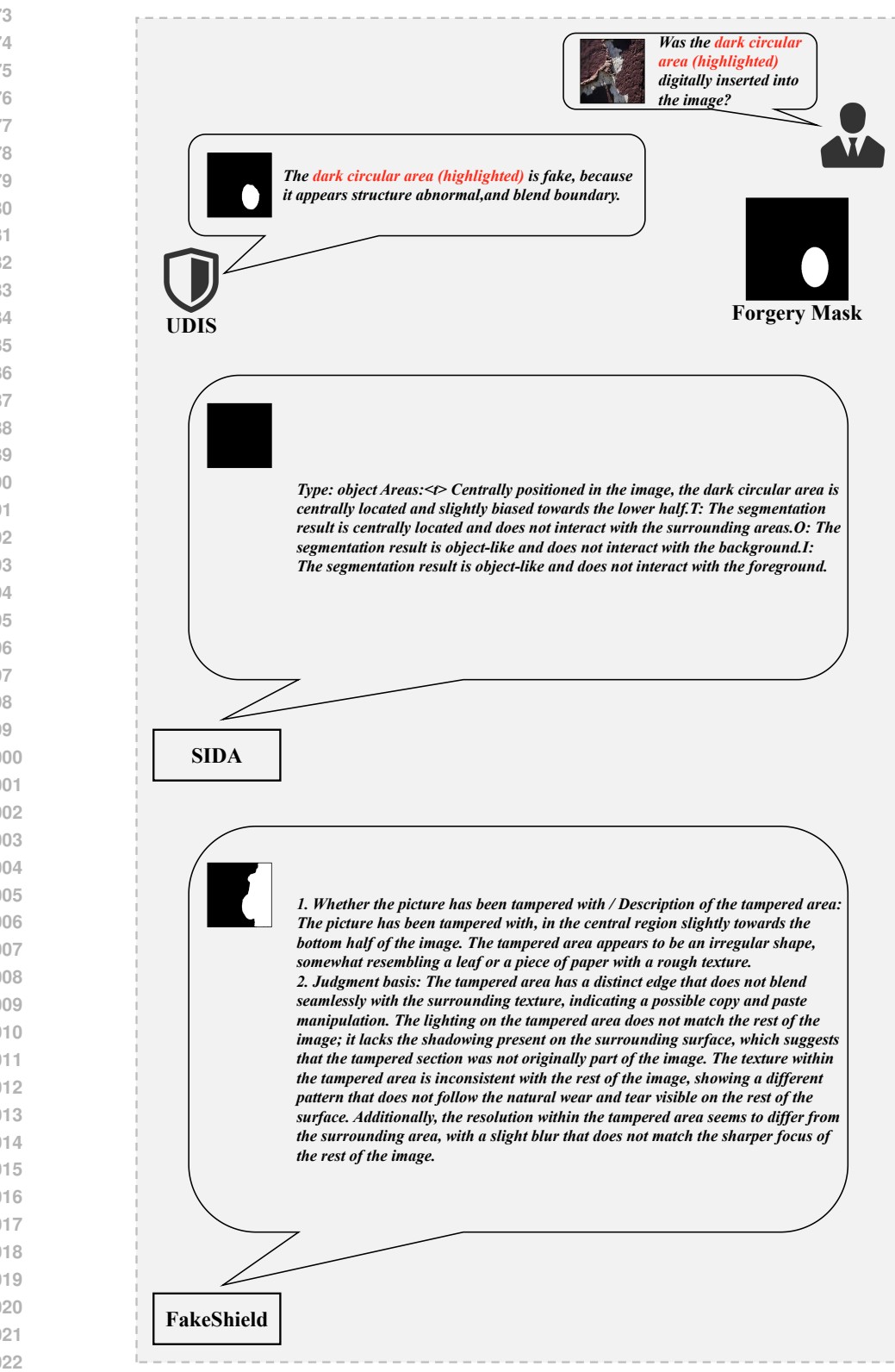

Figure 13: Comparison of forgery localization and interpretability results among different MLLM-based methods. (query related to tampered region)

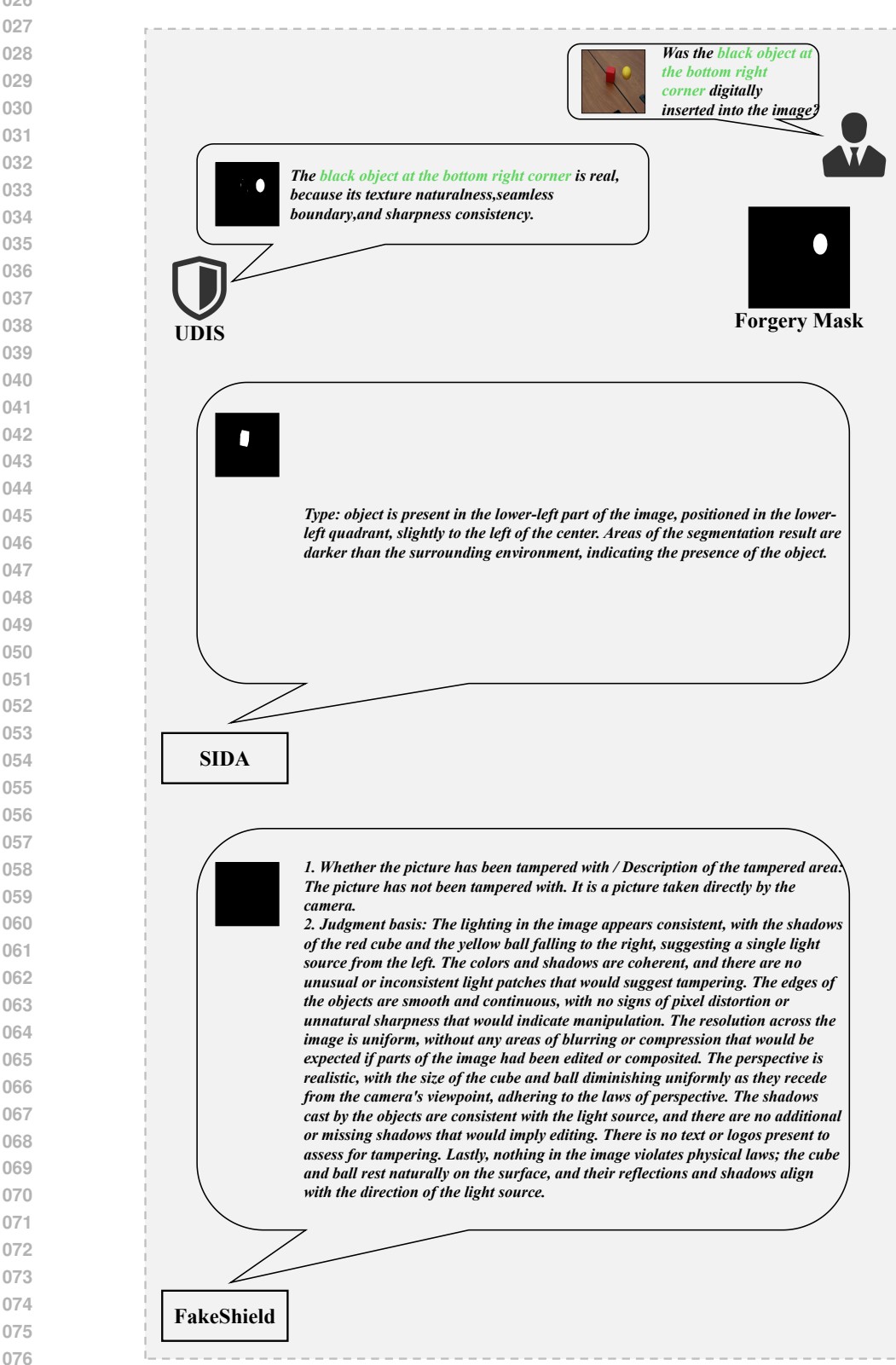

Figure 14: Comparison of forgery localization and interpretability results among different MLLM-based methods (query related to pristine region).

