# OpenReview forum: "UDIS: A User-query Driven Framework for Image Forgery Localization"
_ICLR.cc/2026/Conference — Submitted to ICLR 2026_

### Official Review · Reviewer_pXs9 · 2025-10-21

**Soundness:** 3
**Presentation:** 3
**Contribution:** 3
**Rating:** 6
**Confidence:** 4

**Summary:**

The paper proposes UDIS, a framework designed to enhance multimodal alignment in MLLM-based Image-For-Language  systems by introducing the principle that interpretability should be regional user query driven. UDIS integrates two key modules: the Query-Guided Module (QGM), which refines visual feature selection based on user queries, and the Evidence-Aware Module, which aligns outputs with visual evidence. The experiments on multiple datasets verify the effectiveness of the proposed UDIS.

**Strengths:**

User-query based IFL is interesting and study-worthy
The experiment results are promosing.

**Weaknesses:**

1, In  query-guided module, is it necessary to introduce two sets of learnable queries, i.e. the query token [QUERY] and the learnable query vectors Q^L? I understand that the authors attempt to capture both global and local feature, what if introduce only one set of learnable tokens, their CA outputs and their average vector as the local and global information? This would be streamline the framework.

2, there is a lack visualization of selected Patch embeddings to verify that the selected patches are indeed related to the query text.

3, How to determine that a query does not refer to any specific region (at the end of subsection 3.3)?

4,  How to acquire T to supervise the  LLM output \hat{T}  in L_{txt}?

5, There is a lack of quantitative study of the methods under object-specified query

6,  During the main experiments, a general query text is adopted for fair comparison, how much performance gain would acuqire if user the forgery region focused query text?


minor questions:
1, what's the final evidence classification accuracy?

2, which layer is the ending of encoding the [QUERY] [LOC] [EVI]?  The last LLM layer?

3, Inference time should be compared.

**Questions:**

see weaknesses

---

> ### Author Response · Authors · 2025-11-20
> **Response to Reviewer pXs9 (Part 1)**
>
> Thank you for your thoughtful feedback, which has been crucial in helping us improve the quality of our work. We have carefully revised the manuscript accordingly and are open to continuing the discussion.
>
> > Weakness \#1
>
> The introduction of $\texttt{[QUERY]}$ token is **necessary**.
>
> - The $\texttt{[QUERY]}$ token and the $Q^L$ carry information beyond just global and local differences. More importantly, the $\texttt{[QUERY]}$ token acts as a bridge between MLLM and QGM, enabling MLLM to learn which image regions are of primary concern in the user's problem—namely, visual-text alignment at the input—a capability that $Q^L$ cannot achieve.
>
> > Weakness \#2
>
> - We visualize the visual attention regions of UDIS in Figure 4. The results in the third column show that the patch embeddings selected by UDIS are highly relevant to the user query.
> - The relevant description is in lines 472-474, specifically stating, "It can be observed that the design of QGM effectively concentrates the model’s attention on the regions most relevant to the user query, thereby improving both the accuracy of the responses and the IFL capability." This expression is indeed not clear enough, and we will revise it to: "It can be observed that the selected patch embeddings of QGM are indeed related to the user query, thereby focusing the model’s attention on user-specific regions and improving both the accuracy of the responses and the IFL capability."
> - In our revised submission, we compared the visual attention regions of UDIS when different questions were asked for the same image, and the corresponding adjustments were observed, further supporting the claim that "the selected patches are indeed related to the query text." (In Appendix A.4.5)
>
> > Weakness \#3
>
> The visual feature filtering process is applied only during the training phase to suppress the influence of irrelevant features, thereby improving learning efficiency and enhancing the model’s input alignment capability. During the testing phase, the model directly relies on the learned alignment ability to identify the image regions of interest to the user, eliminating the need for this additional filtering step.
> - During training phase, we designed several question templates. Some templates include replaceable keywords to simulate scenarios where users ask questions about a specific region of an image, while the others ask questions about the entire image. Templates that ask questions about the entire image are considered "does not refer to any specific region." For example, (1) "Please find 'the forged region(s)'." and (2) "Is the {Region} a manipulated or edited element within this image?". We have added the template design details in the appendix (Figure 10).
> - During testing, no judgment is made regarding whether the user’s question involves a specific region. Given that the ability to capture the corresponding image region from the user query—i.e., input-level alignment—has already been learned during training via the $\texttt{[QUERY]}$ token, image feature filtering proves to be of limited effectiveness.
> - We have revised the text accordingly (lines 296-297) to address this point. Thank the reviewers for pointing out the issues.
> - We also compare the localization performance with and without image feature filtering during testing. (the query corresponds to specific regions, e.g., Was the {Region} digitally inserted into the image?) The results indicate that this process has no significant impact on IFL performance. The IoU metrics are reported below.
>
> |               | CASIAv1  | NIST16 | Korus   | AutoSplice | CocoGlide |
> |---------------|----------|--------|---------|------------|-----------|
> | w/o filtering | .593     | .371   | .263    | .503       | .535      |
> | w/ filtering  | .593     | .371   | .263    | .503       | .536      |
>
> > Weakness \#4
>
> - The format of T is "{authenticity},{region},{authenticity evidence}" (e.g., real,bear in the middle center,texture naturalness,sharpness consistency), and the annotation process is as described in Sec. 3.2 in main text.
> - We use Qwen to identify objects in the image or directly use the annotations from the COCO dataset as candidate {region}. We then distinguish the evidence {authenticity evidence} that can be used to determine the authenticity of these {region} according to predefined rules. Finally, we construct T according to the above format.

---

> ### Author Response · Authors · 2025-11-20
> **Response to Reviewer pXs9 (Part 2)**
>
> > Weakness \#5 and \#6
>
> We thank the reviewers for pointing out the issues, and we have supplemented this experiment.
>
> - The IoU metric are reported below. The experimental results show that object-specified query at test time would not affect the localization performance of UDIS.
> - The idea of improving the localization ability of forensic MLLM by obtaining information from user queries is mainly reflected in the training process, and related ablation experiments have also proven its effectiveness (Table 3 (2) and (6), F1 improvement of 8.6%, IoU improvement of 7.6%). During the testing process, UDIS mainly relies on its learned forensic knowledge to complete the localization task, rather than whether the user's question mentions the tampered regions.
>
> | Question              | CASIAv1  | NIST16 | Korus   | AutoSplice | CocoGlide |
> |-----------------------|----------|--------|---------|------------|-----------|
> | About Pristine Region | .593     | .371   | .263    | .503       | .535      |
> | About Tampered Region | .593     | .371   | .263    | .503       | .536      |
> | Without Any Region    | .593     | .371   | .263    | .503       | .535      |
>
> > Minor Question \#1
>
> - We tested the evidence classification accuracy on AutoSplice, and the results are below. Considering that authenticity evidence classification is a relatively complex multi-class task, a classification accuracy of 77.6% is not high, but it demonstrates that UDIS has a certain level of judgment ability.
>
> | Method | Acc  | F1   |
> |--------|------|------|
> | UDIS   | .776 | .863 |
>
> > Minor Question \#2
>
> - $\texttt{[QUERY]}$ $\texttt{[LOC]}$ $\texttt{[EVI]}$ are extracted from the output of the last layer of the LLM.
>
> > Minor Question \#3
>
> - The inference time of different methods are below. Compared to other methods, our proposed scheme achieves relatively fast inference efficiency.
>
> | Methods   | Time/ms |
> |-----------|---------|
> | MVSS-Net  | 146.81  |
> | IML-ViT   | 91.78   |
> | APSC-Net  | 143.13  |
> | SparseViT | 13.70   |
> | Mesorch   | 85.75   |
> | SIDA      | 165.97  |
> | UDIS      | 89.65   |

---

### Official Review · Reviewer_c3nf · 2025-10-29

**Soundness:** 3
**Presentation:** 3
**Contribution:** 3
**Rating:** 8
**Confidence:** 4

**Summary:**

The article proposes the interesting concept of user-query driven and designs an image forgery localization framework UDIS based on a multimodal large language model. It consists of two modules: Query-Guided Module and Evidence-Aware Module, which respectively enhance the visual-text alignment capabilities of the forensic large model at the input and output ends, solving the defect that the forensic large model cannot answer specific user questions related to forensics and further improving the localization capability.

**Strengths:**

1. The article proposes a user-query driven design principle for forensic models, which offers valuable insights and practical value.
2. The proposed QGM guides the model's focus on specific image regions by selecting image features most relevant to the user's question, improving the model's visual-text alignment capabilities at the input-level. The EAM, through the auxiliary task of authenticity evidence classification, transforms knowledge from text annotations into localization capability, thereby enhancing the model's visual-text alignment capabilities at the output-level. The design motivations for both modules are clearly stated and the structure is simple, offering practical value.
3. Compared to state-of-the-art image forgery localization methods, the proposed method achieves superior performance.
4. The article is clearly written and well-organized.

**Weaknesses:**

1. It is recommended to provide the probability distribution of different authenticity evidence on the training dataset to verify that the definitions of these authenticity evidence have sufficient ability to distinguish different images.
2. More experiments are needed to verify the effectiveness of the method, such as the localization ability on AI-edited image datasets such as GRE[1].

[1] Rethinking Image Editing Detection in the Era of Generative AI Revolution

**Questions:**

Please refer to the weaknesses.

---

> ### Author Response · Authors · 2025-11-20
> **Response to Reviewer c3nf**
>
> We are grateful for your thoughtful review. We have strived to address every concern you raised and look forward to your further discussion.
>
> > Weakness \#1
>
> - We thank the reviewer for pointing out the issues, and the distribution of different types of authenticity evidence are as follows:
>
> | Pristine Evidence | Color Consistency | Texture Naturalness | Sharpness Consistency | Seamless Boundary | Frequency Consistency |
> |-------------------|-------------------|---------------------|-----------------------|-------------------|-----------------------|
> |                   | 23.35%            | 78.00%              | 43.62%                | 62.88%            | 21.15%                |
>
> | Forgery Evidence  |   Color Difference | Blur&nbsp;&nbsp;&nbsp;&nbsp;&nbsp;&nbsp;&nbsp;&nbsp;&nbsp;&nbsp;&nbsp;&nbsp;&nbsp;&nbsp;&nbsp;&nbsp;&nbsp;&nbsp;&nbsp;&nbsp;&nbsp;&nbsp;&nbsp;&nbsp;&nbsp;&nbsp; |     Structure Abnormal |   Texture Abnormal |         Blend Boundary |
> |-------------------|-------------------|---------------------|-----------------------|-------------------|-----------------------|
> |                   |             84.45% |               28.09% |                 88.44% |             30.23% |                 93.00% |
>
> > Weakness \#2
>
> - The test results on the GRE are shown in the table below. Because the GRE uses more advanced AI editing algorithms, which are more deceptive, most methods performed poorly in locating the tampered areas. However, UDIS still achieved the best results at this stage.
>
> | Methods   | F1       | IoU      |
> |-----------|----------|----------|
> | MVSS-Net  | .022     | .014     |
> | IML-ViT   | .064     | .039     |
> | APSC-Net  | .040     | .028     |
> | Mesorch   | .038     | .027     |
> | SparseViT | .041     | .027     |
> | SIDA      | .080     | .052     |
> | UDIS      | **.102** | **.066** |

---

### Official Review · Reviewer_DR1U · 2025-10-29

**Soundness:** 3
**Presentation:** 3
**Contribution:** 3
**Rating:** 8
**Confidence:** 3

**Summary:**

This paper addresses a key limitation in existing MLLM-based Image Forgery Localization (IFL) methods: their weak visual-text alignment, which results in a failure to focus on specific regions mentioned in user queries. The authors propose a paradigm shift from a "global outcome driven" approach to a "regional user-query driven" one.

To enable this, they introduce UDIS (User-query Driven Image Shield), a new framework with two core components: 1) a Query-Guided Module (QGM) to enhance input-level alignment by filtering visual features based on the query, forcing the MLLM to attend to relevant regions ; and 2) an Evidence-Aware Module (EAM) to improve output-level alignment by using an auxiliary classification task to bridge the gap between coarse-grained textual explanations and fine-grained localization masks.

**Strengths:**

- The design of the EAM identifies a real-world problem: the "discrepancy between the supervisory signals" (coarse, structured text vs. fine-grained, unstructured masks). Using "authenticity evidence" (e.g., "Color Difference," "Blend Boundary") as a shared, intermediate representation to bridge this modality gap via an auxiliary task  is an intelligent and effective design choice.

- The model achieves state-of-the-art IFL performance (F1, IoU) against numerous baselines, including other MLLM-based methods, across a wide range of test datasets.

**Weaknesses:**

- The process for annotating the forgery evidence (Color Difference, Blur, etc. ) is less clear than the process for the pristine evidence. Appendix A.2  provides concrete, heuristic-based rules for the 5 pristine types (e.g., "compute mean and variance differences in Lab color space"). However, it's not specified if the 5 forgery types are also annotated heuristically or if this was a massive manual annotation effort. If heuristic, this is a crucial detail for reproducibility and understanding potential biases (i.e., is the EAM learning forensics or just learning to replicate the heuristics?).

- he EAM relies on a fixed, discrete taxonomy of 5 forgery and 5 pristine evidence types . This feels somewhat restrictive and similar to older, handcrafted-feature-based methods. It's unclear how this framework would handle novel generative forgeries (e.g., diffusion-based inpainting, as in the CocoGlide dataset ) where the "evidence" might be semantic or stylistic inconsistency, rather than fitting neatly into "Blur" or "Texture Abnormal." A discussion on the EAM's extensibility or its performance on forgeries that defy this taxonomy would be welcome.

**Questions:**

Regarding "Weakness #4," how does the EAM's fixed evidence taxonomy handle modern generative forgeries, such as those in the CocoGlide dataset? Does the model learn to map semantic artifacts (which are not in the list) to one of the existing 5 forgery types, or does the EAM's contribution diminish for these more advanced forgeries?

---

> ### Author Response · Authors · 2025-11-20
> **Response to Reviewer DR1U**
>
> We sincerely thank you for your insightful comments and welcome any further questions you may have.
>
> > Weakness \#1
>
> - Both Forgery Evidence and Pristine Evidence are heuristics. The definition of Forgery Evidence is completely consistent with that of FFTG\[1], and we have cited the relevant literature at line 250 of the main text.
> - Heuristic design does introduce some bias, but the experimental results (Tables 3 (4) and (6)) show that it can also bring some performance improvement (F1 improvement of 4.7%, IoU improvement of 4.3%). However, this is not the optimal approach. In future work, we may consider combining manual annotation or LLM annotation to optimize the solution. Thank you to the reviewer for valuable suggestions.
>
> > Weakness \#2 and Question
>
> - EAM is extensible. Based on fine-grained regional visual features and highly regular text annotations, EAM constructs an auxiliary authenticity evidence classification task for learning. In principle, we can efficiently expand the types of forgeries that EAM can cover simply by increasing the number of categories in the classification task.
> - During training, if a sample appears that does not conform to our definition of authentication evidence, its contribution to EAM is ignored. This does affect EAM's ability to identify more sophisticated forgery samples. However, considering the extensibility of EAM, this limitation can be mitigated to some extent by defining more comprehensive authenticity evidence rules. We will consider this further in future work. Thank you for your valuable comments.
>
> We will further elaborate on the above discussion in the revision.
>
> [1] Sun K, Chen S, Yao T, et al. Towards general visual-linguistic face forgery detection[C]//Proceedings of the Computer Vision and Pattern Recognition Conference. 2025: 19576-19586.

---

> > ### Comment · Reviewer_DR1U · 2025-11-26
> >
> > Thanks for the authors' response. Currently I have no more questions, and I will keep my score.

---

> > > ### Author Response · Authors · 2025-11-27
> > > **Thanks for the recognition of Reviewer DR1U**
> > >
> > > We appreciate you maintaining your score and for your valuable feedback, which we will incorporate to further improve our work.

---

### Official Review · Reviewer_CXft · 2025-11-01

**Soundness:** 2
**Presentation:** 3
**Contribution:** 3
**Rating:** 4
**Confidence:** 4

**Summary:**

UDIS (User-query Driven Image Shield) proposes a new paradigm for Image Forgery Localization (IFL) using Multimodal Large Language Models (MLLMs). Existing methods often fail to align visual attention with user queries, leading to irrelevant or global explanations. UDIS redefines interpretability as regional, user-query driven, introducing two modules. Query-Guided Module (QGM) aligns user queries with visual regions via a [QUERY] token and feature filtering. Evidence-Aware Module (EAM): aligns textual authenticity evidence with visual localization via an [EVI] token and auxiliary classification task. A curated dataset with region-specific queries and authenticity evidence supports training. Experiments across six benchmarks show state-of-the-art localization performance and superior interpretability, with robustness against post-processing distortions.

**Strengths:**

1. Conceptual novelty: This paper redefines interpretability in IFL as user-query driven, providing a fresh paradigm beyond outcome-based localization.

2. Strong multimodal design: The QGM and EAM effectively improve input/output alignment, yielding better focus and explainability.

3. Comprehensive dataset curation: This paper Introduces region-level Q&A-style annotations and evidence categories, enhancing real-world relevance.

4. Extensive experiments, ablations, and robustness tests consistently show superior F1/IoU and interpretability metrics.

**Weaknesses:**

1. The result in Figure 9 is quite puzzling: it appears that the model predicts a mask regardless of whether the answer is “real” or “fake”.

2. Moreover, the paper forces the large‑language model (LLM) to produce detection results in a very rigid format: “the xxx is real/fake, because its xxx (Evidence type)”. The first part is a binary classification (real vs. fake) and the second part is a fixed multi‑class evidence type. Such a rigid output format significantly undermines the LLM’s inherent advantage of generating varied and rich textual responses. In effect, the LLM becomes replaceable—another backbone could output the same fixed pattern with similar performance.

3. In Table 2, the model's forgery detection capability is presented, but it is compared mainly against other LLM‑based methods rather than the non‑LLM based approaches (e.g., those listed in Table 1 such as MVSS‑Net, IML‑ViT, APSC‑Net). Furthermore, the reported detection accuracy for the proposed method (UDIS) is only 0.732, which seems relatively low and perhaps insufficient for acceptance in practical forensic settings. Additionally, Table 2 lacks evaluations on multiple benchmarks in the way Table 1 provides coverage across several datasets.

**Questions:**

Please refer to the weakness.

---

> ### Author Response · Authors · 2025-11-20
> **Response to Reviewer CXft**
>
> We appreciate your insightful comments, which we have carefully considered. We believe our revisions have strengthened the paper and welcome further discussion.
>
> > Weakness \#1
>
> The explanatory response and forgery location mask from UDIS are **not contradictory**.
>
> - Our core argument is that forensic MLLM should output a textual explanation based on the user's question.
> - UDIS has two outputs: (1) the authenticity judgment and analysis of the image region mentioned in the user's question, and (2) the forgery location mask that always indicates the tampered regions within the given image.
> - For example, in the third sub-image of Figure 9 (Figure 11 in the revision), the “bridge” is real, and the “boat” is tampered with. The user asked about the authenticity of the “bridge”, and UDIS answered "The bridge is real," and marked the boat with a mask. The two outputs of UDIS are not contradictory.
>
> > Weakness \#2
>
> Structured output facilitates MLLM's learning of forensic knowledge. Understanding and answering user questions is a core function of UDIS, rather than generating rich textual responses. "Another backbone" **cannot replace** the MLLM's role in UDIS.
>
> - Considering that there are two key pieces of information that MLLM is expected to learn during UDIS training: (1) the authenticity of the image region inquired about by the user, and (2) the criteria for judging the authenticity of that region. Structured output facilitates MLLM's learning of these two core forensic knowledge. While other information enriches the output text of MLLM, it is mostly redundant and more likely to lead to a decrease in accuracy.
> - The core function provided by UDIS is: to analyze the authenticity of different regions of an image based on different user questions. This function requires the model to have a certain level of text comprehension ability, which "another backbone" cannot achieve.
>
> > Weakness \#3
>
> - $ACC_{r/f}$ measures the model’s ability to (1) correctly locate the user-mentioned image region, and (2) accurately determine its authenticity.
> - The $ACC_{r/f}$ metric is defined as: $ ACC_{r/f}=\frac{A_{acc}}{A_{hit}}.$ Here, $A_{hit}$ denotes the number of model responses that correctly identify and mention the image region referred to in the user query. Only when the model explicitly grounds the queried region in its answer do we consider the region “hit”. Among these hits, $A_{acc}$ counts the subset of responses that further provide the correct authenticity judgment (real or fake) for that specific region.
> - It is important to note that models such as MVSS-Net only output a binary authenticity prediction for the entire image and unable to determine whether "correctly locate the user-mentioned image region". Consequently, they cannot be evaluated under the same protocol.
> - The $ACC_{r/f}$ of 73.2% is indeed low, but it is still optimal compared to existing methods. The improvement in this metric is partly due to the IFL task, while UDIS has already achieved SOTA performance (Table 1 in the main text). We will further explore better solutions in future work.
> - We have further tested the method on additional datasets, and the proposed approach achieves the best performance in most cases. Although FakeShield achieved the best $ACC_{r/f}$ on the NIST16 and Korus datasets, its relatively low $HR_{query}$ makes it difficult to truly perform well in real-world scenarios.
>
> We thank the reviewer for valuable suggestions.
>
> | $HR_{query}$    | CASIAv1  | NIST16   | Korus    | AutoSplice | CocoGlide |
> |-----------------|----------|----------|----------|------------|-----------|
> | SIDA*           | .608     | .355     | .606     | .664       | .571      |
> | FakeShield*     | .290     | .313     | .265     | .319       | .329      |
> | UDIS w/o QA,QGM | .205     | .110     | .135     | .176       | .243      |
> | UDIS w/o QGM    | .873     | .844     | .835     | **.919**   | **.828**  |
> | UDIS            | **.905** | **.846** | **.877** | **.919**   | .819      |
>
> | $Acc_{r/f}$     | CASIAv1  | NIST16   | Korus    | AutoSplice | CocoGlide |
> |-----------------|----------|----------|----------|------------|-----------|
> | SIDA*           | .295     | .220     | .226     | .126       | .312      |
> | FakeShield*     | .524     | **.731** | **.911** | .521       | .533      |
> | UDIS w/o QA,QGM | .561     | .554     | .741     | .607       | .585      |
> | UDIS w/o QGM    | .683     | .584     | .725     | .717       | **.677**  |
> | UDIS            | **.722** | .640     | .678     | **.741**   | .650      |

---

### Author Response · Authors · 2025-12-03
**Summary of rebuttal and discussions**

We have received a notification from the Program Chairs regarding a recent information breach in the OpenReview system. We have briefly summarized the core contributions of our work and the key reviewer concerns that we addressed during the rebuttal phase. We hope this summary will serve as a helpful reference for the final evaluation.

### **The main contributions of our work**

- We propose a **regional user-query driven principle** to address the issue of MLLM-based forensic models' answers being irrelevant to user questions, and further improve the image forgery localization (IFL) capabilities of these models.
- Based on this principle, we designed a **Query-Guided Module** and an **Evidence-Aware Module** to enhance the alignment capabilities of the MLLM-based IFL models at the input and output levels, respectively.
- Experimental results show that our method significantly outperforms existing state-of-the-art methods (16.2% improvement in F1 score and 11.1% improvement in IoU).

### **Summary of key responses to the reviewers**

**Strengths Recognized Across Reviewers**

- Novel concept definition: All reviewers agreed that shifting IFL from a global outcome–driven paradigm to regional user-query driven is both novel and impactful.
- Clear technical contributions with sound motivation: QGM improves input-level visual–text alignment. EAM bridges the gap between coarse textual supervision and fine-grained masks. Reviewer CXft, DR1U and c3nf widely considered the design intuitive, clean, and practically valuable.
- Strong empirical results and broad evaluation: UDIS achieves SOTA on six benchmarks and maintains superior performance even on challenging AI-edit datasets (e.g., CocoGlide, GRE).
- High clarity and solid presentation: All reviewers gave high scores on presentation and contribution.

**For Reviewer CXft**

> Whether textual explanations conflict with the predicted mask

- **The two outputs of UDIS are not contradictory**. We clarified the dual-output design: (1) the authenticity judgment and analysis of the image region mentioned in the user's question, and (2) the localization mask highlights all manipulated areas.

> Whether structured output limits MLLM expressiveness

- **The structured format is essential for stable forensic supervision**. We explained that there are two key pieces of information that MLLM is expected to learn: (1) the authenticity of the image region inquired about by the user, and (2) the criteria for judging the authenticity of that region, while other information is redundant.

> Whether the MLLM is replaceable in UDIS

- **The MLLM is irreplaceable.** The reason UDIS achieves **better IFL performance** is that it leverages the inherent visual-text alignment capabilities of the MLLM and trains its ability to judge the authenticity of a specific region and provide corresponding explanations. This is something that "another backbone" cannot achieve.

> Fairness of $ACC_{r/f}$ and comparison with non-MLLM methods

- We explained that $ACC_{r/f}$ measures (1) correctly locating the user-mentioned image region and (2) accurately determining its authenticity. The output of the traditional IFL model cannot determine whether "correctly locate the user-mentioned image region", so it cannot be evaluated under the same protocol. We also added multi-dataset comparisons confirming that UDIS achieves SOTA under fair evaluation.

**For Reviewers DR1U, c3nf and pXs9**

We thank the reviewers for their recognition and positive scores. We have responded to all their suggestions and will incorporate them into the final revised manuscript.

We sincerely thank the AC and all reviewers for their time and effort throughout the review process and greatly appreciate their valuable feedback. We have made every effort to address the reviewers’ concerns during the rebuttal phase, and we believe these improvements have substantially strengthened our paper.

---

### Meta-Review · Area_Chair_vAvs · 2026-01-05

**Summary:**

- CXft, DR1U: The rigid, discrete evidence design limits extensibility, weakens the claimed benefits of using an LLM, and raises doubts about applicability to novel generative forgeries.

- DR1U: While pristine evidence annotation is clearly defined via heuristics, the process for annotating the forgery evidence is less clear. This ambiguity leads to concerns about reproducibility and bias.

- CXft, c3nf, pXs9: The experimental section is seen as insufficiently comprehensive, lacking strong baselines, broader benchmarks, and efficiency evaluation.

**Reviewer Concerns:**

- The rigid output issue remains unaddressed. The rebuttal does not provide technical or empirical evidence demonstrating that the LLM’s generative or reasoning capabilities are essential under the imposed structured output constraint. Moreover, the rebuttal does not explain whether a non-LLM backbone could achieve similar functionality.

- The issue of forgery evidence annotation ambiguity is well addressed in the rebuttal, which sufficiently clarifies the annotation process and alleviates concerns regarding reproducibility and bias.

- The issue of experimental evaluation is only partially addressed. The authors clarify the evaluation protocol and add results on additional datasets, which address concerns about benchmark coverage. However, the low absolute performance and the lack of convincing justification for practical forensic applicability remain unresolved.

**Reviewer Scores:**

All reviewers are likely to maintain their ratings.

---

### Decision · Program_Chairs · 2026-01-26

Reject